# Ensemble Learned Bloom Filters: Two Oracles are Better than One

**Ming Lin** [1]  **Lin Chen** [2]

## Abstract

Bloom filters (BF) are space-efficient probabilistic data structures for approximate membership testing. Boosted by the proliferation of machine learning, learned Bloom filters (LBF) were recently proposed by augmenting the canonical BFs with a learned oracle as a pre-filter, the size of which is crucial to the compactness of the overall system. In this paper, inspired by ensemble learning, we depart from the state-of-the-art single-oracle LBF structure by demonstrating that, by leveraging multiple learning oracles of smaller size and carefully optimizing the accompanied backup filters, we can significantly boost the performance of LBF under the same space budget. We then design and optimize ensemble learned Bloom filters for mutually independent and correlated learning oracles respectively. We also empirically demonstrate the performance improvement of our propositions under three practical data analysis tasks.

## 1. Introduction

Originally proposed by and named after Bloom in the seminal paper (Bloom, 1970), Bloom filters (BF) are space-efficient data structures for solving the membership lookup problem by compactly recording a given set $\mathcal{I}$ of data items in a bit array and outputting a binary answer when queried. As a structural property, the space compactness of BFs is achieved at the price of limited false positives but no false negative. Due to the neat structure with profound theoretic and practical implications, BFs have received significant research attention, leading to a palette of extensions and variants including Bloomier filters (Chazelle et al., 2004), counting BF (Bonomi et al., 2006), and invertible Bloom

lookup tables (Goodrich & Mitzenmacher, 2011) etc.

Boosted by the proliferation of machine learning, learned Bloom filters (LBF) were recently proposed (Kraska et al., 2018; Mitzenmacher, 2018), with the design rationale that we can leverage the specific structure exhibited by data items to reduce the space of BFs without sacrificing lookup accuracy. Technically, a pre-filter is constructed by implementing a learning model, termed as the learned oracle or simply oracle for brevity, to filter the queried items. The items with negative responses at the filter are then passed to a backup BF for final check. As the backup filter only needs to deal with filtered items, we can reduce its size and come up with a more compact learning-augmented data structure, even after taking into account the space overhead brought by the oracle. Along this direction, Mitzenmacher (Mitzenmacher, 2018) proposed a sandwich structure to further improve the performance. Under the condition that the score distribution of the oracle is available, several variants attempted to make the best use of the oracle to improve the overall lookup performance: Vaidya *et al.* (Vaidya et al., 2021) segmented the score space of the oracle into multiple regions by using multiple thresholds, and uses separate backup Bloom filters for each region; Sato *et al.* (Sato & Matsui, 2023) accelerated the construction while maintaining the same memory efficiency as Vaidya *et al.* (Vaidya et al., 2021); Dai *et al.* (Dai & Shrivastava, 2020) attributed different number of hash functions in different regions.

In virtually all the state-of-the-art LBF design, the oracle has crucial impact on the performance of the system as a whole. A more accurate oracle results in smaller false positive rate (FPR). However, it has been well-demonstrated (Thompson et al., 2021) that the accuracy of learning models exhibit diminishing performance improvement w.r.t. the model size. Therefore, the size of the learning model may become a potential bottleneck of the LBF if the overall space budget is limited. To quantitatively demonstrate this point, we build an LBF for identifying malicious URLs from 200,000 URLs with a backup BF of 500K bits. We tune the size of the oracle to vary the FPR, as illustrated in Figure 1. We can observe that the size of the oracle occupies non-negligible part of the total memory space and that the accuracy improvement of the oracle paces down significantly with the increase of its size.

The above observation resonates with the technical diffi-

---

[1]School of Computer Science and Engineering, Sun Yat-sen University, Guangzhou, China. [2]Engineering Research Centre of Applied Technology on Machine Translation and Artificial Intelligence, Macao Polytechnic University, Macao SAR, China. Correspondence to: Lin Chen <lchen@mpu.edu.mo>.

*Proceedings of the 42$^{nd}$ International Conference on Machine Learning*, Vancouver, Canada. PMLR 267, 2025. Copyright 2025 by the author(s).

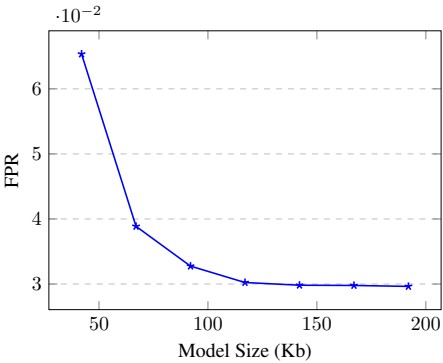

*Figure 1.* FPR of LBFs with an oracle under different sizes and a BF of 500Kbits to detect malicious URLs from 200,000 URLs.

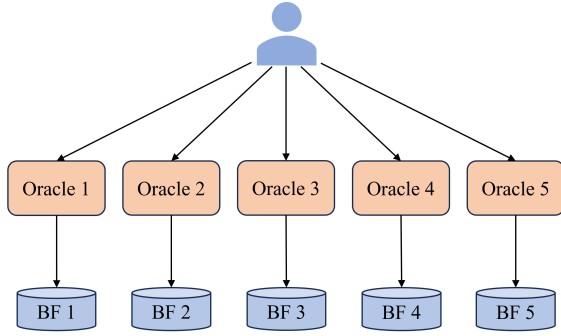

*Figure 2.* Overview of ELBF.

culty and the cost of building strong learning models in the generic machine learning context. One of the most cost-effective technique to boost the accuracy of learning models is ensemble learning (Polikar, 2012), which consists of strategically orchestrating a set of weak models usually of smaller size to form a much stronger model. Ensemble learning has demonstrated remarkable improvements in a variety of learning tasks (Gomes et al., 2017; Mendes-Moreira et al., 2012; Vanerio & Casas, 2017).

Inspired by the ensemble-based learning paradigm, we depart from the state-of-the-art single-oracle structure of the LBF design to investigate the following research question.

> **Q**: How to combine multiple smaller oracles to construct an ensemble learned BF (ELBF) to optimize the overall accuracy under given memory budget? More specifically, given a pool of oracles, which subset of them to choose and how to orchestrate them with backup filters to minimize the overall FPR?

In this paper, we give affirmative answer the question. We demonstrate that, by leveraging multiple learning oracles of smaller size and carefully optimizing the accompanied backup BFs, we can significantly boost the performance of the LBF under the same space budget. We technically achieve this by proposing a new LBF design termed as ensemble learned BFs (ELBF), as depicted in Figure 2. We formulate the optimal design of ELBF as a combinatorial optimization problem: given a pool of oracles and the total space budget, the problem is to choose a subset of oracles and compute the sizes of the backup BFs so that the overall FPR is minimized. We reveal the structural analogy between our problem to the Knapsack problem and develop a Knapsack-based approximate algorithm achieving proven $(\epsilon, \delta)$-optimality with tunable parameters $\epsilon$ and $\delta$ trading off complexity against optimality. In the case of oracles are correlated, we propose a generalized design, ELBF++, letting multiple correlated oracles share a common backup filter. We empirically evaluate our approaches against existing

works under practical data analysis tasks with real-world datasets. Our experiments demonstrate significant performance improvement of ELBFs against the state-of-the-art LBF and its variants.

## 2. Technical Preliminaries

In this section, we give brief introduction on the standard BFs and LBFs, both of which solve the membership lookup problem, i.e., given a set $\mathcal{I}$, outputting a binary answer whether a data item $e$ belongs to $\mathcal{I}$. Throughout the paper, we use calligraphic letters (e.g., $\mathcal{X}$) to denote sets and boldface letters (e.g., $\mathbf{x}$) to denote vectors. Table 1 summarizes major notations.

### 2.1. Canonical Bloom Filters

BFs are space-efficient probabilistic data structures representing a set $\mathcal{I}$. Physically, a BF contains an $m$-bit array $B$ and a family of $k$ independent hash functions denoted as $\mathcal{H} \triangleq \{h_i\}_{i=1}^k$, where $h_i(e)$ maps an item to a random bit in $B$. Each bit of $B$ is initialized to $0$. When inserting an item $e$, we set $B[h_i(e)]$ to $1$ for $1 \le i \le k$. To check whether an item $e \in \mathcal{I}$, we return true if $h_i(e) = 1$ for $1 \le i \le k$, otherwise we return false.

It is clear from the construction of BFs that they do not produce false negatives. However, due to hash collisions, a BF may wrongly return true for an item $e \notin \mathcal{I}$, causing a false positive. Let $n$ denote the number of items recorded in $B$, the probability that a specific bit of $B$ is $0$ is given by $(1 - 1/m)^{kn} \approx e^{-kn/m}$. Thus the probability of a false positive can be computed as $(1 - e^{-kn/m})^k$.

Algebraically, the false positive rate is minimized when $k$ equals $\frac{m}{n} \ln 2$. The minimal value can be computed as $\alpha^{m/n}$, where $\alpha \approx 0.6185$ (Broder & Mitzenmacher, 2004).

### 2.2. Learned Bloom Filters

LBFs exploit a learned model, termed as a learned oracle $O$, briefly denoted as oracle, to perform pre-filtering. If

*Table 1.* Major notations.

| Section 3: Ensemble Learned Bloom Filters with Independent Oracles | | | |
|---|---|---|---|
| $\mathcal{I}$ | item set | $n$ | number of items in $\mathcal{I}$ |
| $\mathcal{H}$ | set of Hash functions | $\alpha$ | 0.6185 |
| $F_{P,i}$ | false positive rate of oracle $i$ | $F_{N,i}$ | false negative rate of oracle $i$ |
| $b_i n$ | number of bits allocated to backup BF $i$ | $\mathcal{O}$ | set of oracles |
| $n_0$ | number of oracles in $\mathcal{O}$ | $L_i n$ | size of oracle $i$ in number of bits |
| $\mathcal{A}$ | set of chosen oracles | $y_i$ | $b_i/F_{N,i}$ |
| $\mathcal{L}(\cdot)$ | Lagrangian function | $\lambda$ | Lagrange multiplier |
| $c$ | auxiliary variable | $F^{(i)}$ | false positive rate of branch $i$ |
| $Tn$ | total space budget in number of bits | $\Delta$ | discretization stepsize of $c$ |
| $\theta_1, \theta_2$ | scaling factors | $v_i$ | value of object $i$ |
| $w_i$ | weight of object $i$ | $\hat{v}_i$ | discretized value of $v_i$ |
| $\hat{w}_i$ | discretized value of $w_i$ | $\hat{v}$ | sum of $\hat{v}_i$ |
| Section 4: Ensemble Learned Bloom Filters with Correlated Oracles | | | |
| $\mathcal{P}(\mathcal{S})$ | a partition of $\mathcal{S}$ | $\Pi(\mathcal{S})$ | set of all possible partitions of $\mathcal{S}$ |
| $g$ | group of oracles | $\Gamma$ | auxiliary learned model |

the learned oracle replies positively, the queried item is considered as a member of $\mathcal{I}$. Otherwise, a backup BF is then used to perform further check as in standard BFs.

Suppose that the oracle takes $Ln$ bits and the backup BF takes $bn$ bits. Denote $F_P$ and $F_N$ as the false positive rate and false negative rate of the oracle. The backup BF only needs to hold $F_N \cdot n$ keys. Hence, the overall false positive rate for the LBF can be derived as

$$F_P + (1 - F_P)\alpha^{b/F_N}. \tag{1}$$

(Mitzenmacher, 2018) also shows an estimation on what size the learning model is required for an improvement for performance. Compared with a standard BF of the same total space, whose false positive rate is $\alpha^{b+L}$, the LBF makes an improvement in the following situations

$$F_P + (1 - F_P)\alpha^{b/F_N} \leq \alpha^{b+L},$$

which translates to the following condition on the size of the learning model:

$$L \leq \log_\alpha \left( F_P + (1 - F_P)\alpha^{b/F_N} \right) - b. \tag{2}$$

# 3. Ensemble Learned Bloom Filters with Independent Oracles

This section presents our design of Bloom filters augmented by ensemble learning. Inspired by ensemble learning, our key idea is to orchestrate multiple learning oracles of small size to replace the learning oracle in the LBF to obtain superior overall performance. In what follows, we first present the ELBF design and then its optimization.

## 3.1. Design

Our ELBFs mobilize multiple learning oracles. To begin with, we assume the oracles are mutually independent, e.g., by learning mutually independent features. Each oracle is backed up by a backup BF, which stores the false negatives produced by the oracle. When an item $e$ is queried, the ELBF replies positively if all the oracles return positively or the oracles return negatively but their corresponding backup BFs return positively, otherwise it replies negatively.

It follows straightforwardly from its construction that our ELBFs do not have false negative, which is inline with the standard BFs. We now analyze its false positive rate. Let $F_{P,i}$ and $F_{N,i}$ denote the false positive and false negative rate of oracle $i$. Suppose $b_i m$ bits are allocated to the backup BF $i$. It follows that backup BF $i$ needs to hold $F_{N,i} \cdot n$ items, with the corresponding optimal false positive rate $\alpha^{b_i/F_{N,i}}$. Thus, the false positive rate for the branch $i$, i.e., the oracle $i$ plus the backup BF $i$, is $F_{P,i}+(1-F_{P,i})\alpha^{b_i/F_{N,i}}$. Therefore, let $\mathcal{O}$ denotes the set of oracles used in the ELBF, which are assumed to be mutually independent. We create a dumb oracle with size 0, FPR $\rightarrow$ 0 and FNR 1, add it into $\mathcal{O}$ to include the degenerated case where adding an oracle will have negative impact on the overall performance. The overall FPR can be computed as follows:

$$\prod_{i \in \mathcal{O}} \left( F_{P,i} + (1 - F_{P,i})\alpha^{b_i/F_{N,i}} \right). \tag{3}$$

## 3.2. Optimization

In this subsection, we optimize the design of ELBF by solving the following minimization problem.

*Problem* 1. Suppose we dispose a set $\mathcal{O}$ of independent oracles, from which we can choose to build our ELBF. Given a total space budget $T \cdot n$ bits, the optimal design of ELBF

computes a subset of oracles $\mathcal{A} \subseteq \mathcal{O}$ to choose and the sizes of their backup BFs $\mathbf{b} \triangleq \{b_i\}_{i \in \mathcal{A}}$ so that the overall false positive rate is minimized. The problem is mathematically formalized as follows.

$$\min_{\mathcal{A} \subseteq \mathcal{O}} \min_{\mathbf{b}} \quad \prod_{i \in \mathcal{A}} \left( F_{P,i} + (1 - F_{P,i})\alpha^{b_i/F_{N,i}} \right)$$
$$\text{subject to:} \quad \sum_{i \in \mathcal{A}} (L_i + b_i) \leq T \tag{4}$$
$$b_i \geq 0, \ \forall i \in \mathcal{A}.$$

Consider the inner minimization problem for a given $\mathcal{A}$. Denote $y_i \triangleq b_i/F_{N,i}$. We can rewrite the inner minimization problem as below.

$$\min_{\mathbf{b}} \quad \sum_{i \in \mathcal{A}} \ln \left( F_{P,i} + (1 - F_{P,i})\alpha^{y_i} \right)$$
$$\text{subject to} \quad \sum_{i \in \mathcal{A}} (L_i + y_i F_{N,i}) \leq T \tag{5}$$
$$y_i \geq 0, \ \forall i \in \mathcal{A}.$$

Using augmented Lagrangian approach, we write the following Lagrangian function for optimization:

$$\mathcal{L}(y_1, y_2, \ldots, y_n, \lambda) = \sum_{i \in \mathcal{A}} \ln \left( F_{P,i} + (1 - F_{P,i})\alpha^{y_i} \right)$$
$$- \lambda \left( \sum_{i \in \mathcal{A}} (L_i + y_i F_{N,i}) - T \right), \tag{6}$$

where $\lambda \leq 0$ is the Lagrange multiplier. The partial derivatives with respect to $y_i$ and $\lambda$ can be derived as below.

$$\begin{cases} \dfrac{\partial \mathcal{L}}{\partial y_i} = \dfrac{(1 - F_{P,i})\alpha^{y_i} \ln \alpha}{F_{P,i} + (1 - F_{P,i})\alpha^{y_i}} - F_{N,i}\lambda \\ \dfrac{\partial \mathcal{L}}{\partial \lambda} = T - \sum_{i \in \mathcal{A}} (L_i + y_i F_{N,i}) \end{cases} \tag{7}$$

We observe that the partial derivative for $y_i$ equals 0 when

$$\frac{(1 - F_{P,i})\alpha^{y_i}}{\left( F_{P,i} + (1 - F_{P,i})\alpha^{y_i} \right) F_{N,i}} = \frac{\lambda}{\ln \alpha} \triangleq c, \forall i \in \mathcal{A}. \tag{8}$$

In the above equation, we define an auxiliary variable $c$, which serves as a pivot in our analysis. Armed with $c$, we can derive the optimal $b_i^*$ for branch $i$ as follows.

$$b_i^* = F_{N,i} \log_\alpha \frac{c F_{N,i} F_{P,i}}{(1 - c F_{N,i})(1 - F_{P,i})}, \tag{9}$$

from which we further get the minimal false positive rate for branch $i$, denoted by $F^{(i)}$, as

$$F^{(i)} = F_{P,i} + (1 - F_{P,i})\alpha^{b_i/F_{N,i}} = \frac{F_{P,i}}{1 - c F_{N,i}} \tag{10}$$

Noticing that $b_i \geq 0$ and $F^{(i)} \in (0, 1]$, we can bound $c$ by $0 < c \leq \frac{1 - F_{P,i}}{F_{N,i}}$.

Armed with the above analysis, we now proceed to consider the outer minimization problem. To this end, we define a set of binary variables $\{x_i\}_{i \in \mathcal{O}}$ to indicate whether we choose oracle $i$. After some straightforward algebraic operations, we rewrite the outer minimization problem as follows.

$$\max_{\mathbf{x}} \quad \sum_{i \in \mathcal{O}} x_i \cdot \ln \frac{1 - c F_{N,i}}{F_{P,i}}$$
$$\text{subject to} \quad \sum_{i \in \mathcal{O}} x_i \cdot (L_i + b_i^*) \leq T \tag{11}$$
$$x_i \in \{0, 1\}, \ \forall i \in \mathcal{O}.$$

By casting each oracle $i$ to an object with value $\ln \frac{1 - c F_{N,i}}{F_{P,i}}$ and weight $L_i + b_i^*$, we observe that our problem has elegant analogy to the famous Knapsack problem, which is known to be NP-hard (Garey & Johnson, 1979). It then follows that our problem is also NP-hard. However, being weakly NP-hard, the Knapsack problem with integer weights can be solved by dynamic programming in pseudo-polynomial time. Suppose we can invoke an oracle, termed as KNAPSACK, that solves the canonical Knapsack problem with integer weights and values. We next develop a Knapsack-based algorithm augmented by the scaling technique to find an $(\epsilon, \delta)$-optimal solution of our problem, as formalized below. The rationale behind is to allow a small quantity of overflow on the space budget to achieve near-optimal FPR.

**Definition 3.1.** The configuration of an ELBF is called $(\epsilon, \delta)$-optimal if its FPR is within $\epsilon$ to the optimal and its space is upper-bounded by $(1 + \delta)T$.

---

**Algorithm 1** ELBF optimization

---
**Input:** oracles set $\mathcal{O}$, space budget $T$, scaling factors $\theta_1, \theta_2$, stepsize $\Delta$
**Output:** selected oracle set $\mathcal{A}$, backup BF setting $\mathbf{b}$
1: **Initialization:** $\hat{v}^* \leftarrow 0, \mathcal{A}^* \leftarrow \emptyset, c \leftarrow 0$
2: **while** $0 \leq c \leq \max\limits_{i \in \mathcal{O}} \dfrac{1 - F_{P,i}}{F_{N,i}}$ **do**
3:   $c \leftarrow c + \Delta$
4:   **for** each $j \in \mathcal{O}$ **do**
5:    $\hat{v}_j \leftarrow \max \left\{ 0, \left\lceil \theta_1 \cdot \ln \frac{1 - c F_{N,j}}{F_{P,j}} \right\rceil \right\}$
6:    $\hat{w}_j \leftarrow \left\lfloor \theta_2 \left( L_j + F_{N,j} \log_\alpha \frac{c F_{N,j} F_{P,j}}{(1 - c F_{N,j})(1 - F_{P,j})} \right) \right\rfloor$
7:   **end for**
8:   $(\hat{v}, \mathcal{A}) \leftarrow \text{KNAPSACK}(\mathcal{O}, \hat{\mathbf{v}}, \hat{\mathbf{w}}, \lceil \theta_2 T \rceil)$
9:   **if** $\hat{v} > \hat{v}^*$ **then**
10:    $\hat{v}^* \leftarrow \hat{v}, \mathcal{A}^* \leftarrow \mathcal{A}$
11:    $b_i^* \leftarrow F_{N,i} \log_\alpha \frac{c F_{N,i} F_{P,i}}{(1 - c F_{N,i})(1 - F_{P,i})}, \forall i \in \mathcal{A}$
12:   **end if**
13: **end while**

---

The pseudo-code of our algorithm is given in Algorithm 1. The core part of our algorithm is the **while** loop, in which we scan $c$ with stepsize $\Delta$. For each discretized value of $c$, i.e., $i\Delta$, we scale the values $v_j$ and the weights $w_j$ for each oracle $j \in \mathcal{O}$. We then invoke the Knapsack algorithm by regarding each oracle as an object with the total weight constraint scaled up to $\theta_2 T$. Our algorithm returns the optimal Knapsack solution among all discretized $c$, which is also an $(\epsilon, \delta)$-optimal ELBF configuration, as presented in Theorem 3.2 and proved in Appendix A.

**Theorem 3.2.** *Under the condition* $\theta_1 \geq \frac{n_0}{\epsilon}$ *and* $\theta_2 \geq \frac{n_0+1}{T\delta}$, *Algorithm 1 outputs an* $(\epsilon, \delta)$-*optimal ELBF configuration.*

We conclude this section by analyzing the time complexity of our algorithm. The complexity to calculate the values and weights and solve a specific Knapsack problem is $O(n_0 + n_0\lceil\theta_2 T\rceil) = O(n_0^2/\delta)$. As the above operations are performed for each discretized value of $c$, the overall time complexity sums up to $O\left(\max_{j\in\mathcal{O}} \frac{n_0^2(1-F_{P,j})}{F_{N,j}\Delta\delta}\right)$, i.e., asymptotically $O\left(n_0^2/(\delta\Delta)\right)$.

## 4. Ensemble Learned Bloom Filters with Correlated Oracles

In the previous section, the assumption of mutually independent oracles was made. Nevertheless, in real-world scenarios, oracles frequently exhibit correlations. In this section, we address this correlated case by devising a generalized version of ELBF called ELBF++.

The design rationale of ELBF++ lies in allowing multiple correlated oracles to share a common backup filter, as illustrated in Figure 3. This is achieved through carefully selecting a subset of oracles and optimizing the corresponding backup filter parameters. Specifically, ELBF++ operates as follows: Each backup filter stores the false negatives of all the selected oracles connected to it. When an item is queried, ELBF++ returns a positive response if all the selected oracles give positive results or, if some of them return negatively, but their corresponding backup filters return positively. Otherwise, ELBF++ returns a negative response.

Before delving into the design of ELBF++, we note that ELBF can be regarded as a degenerated version of ELBF++. As proved in Appendix, when oracles are mutually independent, attributing a separate backup filter to each oracle leads to strictly lower FPR than letting multiple oracles sharing a common backup filter, which justifies the ELBF design. However, when oracles are correlated, the situation is more complex such that sharing a common backup filter among correlated oracles helps reduce FPR.

The central research problem in the design of ELBF++ is to choose the appropriate set of correlated oracles for each backup filter. To formalize the problem, suppose we dispose

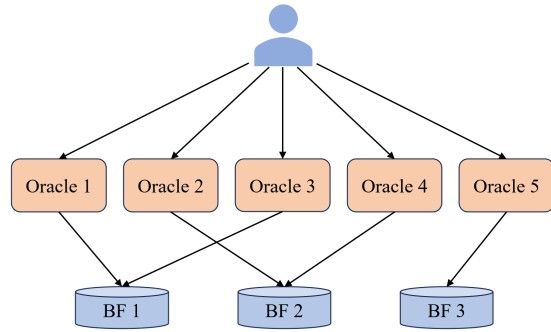

*Figure 3.* Overview of ELBF++.

a set $\mathcal{O}$ of oracles, some of which are correlated. Let $\mathcal{S}$ denote a subset of $\mathcal{O}$ and $\mathcal{P}(\mathcal{S})$ denote a partition of $\mathcal{S}$. For each element of $g \in \mathcal{P}(\mathcal{S})$ representing a subset of oracles, we consider them sharing the same backup filter, thereby forming a branch. We define $F_{P,g}$ and $F_{N,g}$ as the FPR and FNR of branch $g$, respectively. Assuming $b_g m$ bits are allocated to the backup filter, the FPR of branch $g$, denoted by $F^{(g)}$, is given by

$$F^{(g)} = F_{P,g} + (1 - F_{P,g})\alpha^{b_g/F_{N,g}}. \qquad (12)$$

For a given partition $\mathcal{P}(\mathcal{S})$ consisting of $m$ branches, we denote $X_i(\mathcal{P}(\mathcal{S}), \mathbf{b})$, concisely denoted by $X_i$ when the context is clear, as the event that branch $i$ returns false positively. The overall FPR is then $\Pr(\bigcap_{i=1}^m X_i)$. Define

$$\Gamma(\mathcal{P}(\mathcal{S}), \mathbf{b}) \triangleq \frac{\Pr\left(\bigcap_{i=1}^m X_i\right)}{\prod_{i=1}^m \Pr(X_i)}.$$

We can express $\Pr\left(\bigcap_{i=1}^m X_i\right)$ as

$$\Pr\left(\bigcap_{i=1}^m X_i\right) = \Gamma(\mathcal{P}(\mathcal{S}), \mathbf{b})\prod_{i=1}^m \Pr(X_i)$$
$$= \Gamma(\mathcal{P}(\mathcal{S}), \mathbf{b}) \prod_{g\in\mathcal{P}(\mathcal{S})} \left(F_{P,g} + (1-F_{P,g})\alpha^{b_g/F_{N,g}}\right). \qquad (13)$$

We can observe that computing the optimal value of $\Pr\left(\bigcap_{i=1}^m X_i\right)$ requires evaluating $\Gamma(\mathcal{P}(\mathcal{S}), \mathbf{b})$ for each partition $\mathcal{P}(\mathcal{S})$ and each possible size configuration of the backup filters $\mathbf{b}$, whose complexity is super-exponential.

To reduce the above complexity, we adopt a learning-based approach. Specifically, we randomly sample a subset of partitions and configurations and then train an auxiliary learned model that, given any partition $\mathcal{P}(\mathcal{S})$ and any size configuration of the backup filters $\mathbf{b}$, estimates $\Pr\left(\bigcap_{i=1}^m X_i\right)$. Existing works have demonstrated the feasibility of learning effective models with small sample size (Loo et al., 2019; Lu et al., 2023; Sendera et al., 2021). In our work, we

leverage the random forest model to learn $\Gamma(\mathcal{P}(\mathcal{S}), \mathbf{b})$. The details of our training algorithm are given in Appendix C.

Suppose our random forest model for learning $\Gamma(\mathcal{P}(\mathcal{S}), \mathbf{b})$ takes $Un$ bits. Let $\Pi(\mathcal{S})$ denote the set of all possible partitions of $\mathcal{S}$. We can formulate the problem of optimal ELBF++ design as follows.

$$\min_{\mathcal{S} \subseteq \mathcal{O}} \min_{\mathcal{P}(\mathcal{S}) \in \Pi(\mathcal{S})} \min_{\mathbf{b}}$$
$$\Gamma(\mathcal{P}(\mathcal{S}), \mathbf{b}) \prod_{g \in \mathcal{P}(\mathcal{S})} \left( F_{P,g} + (1 - F_{P,g})\alpha^{b_g/F_{N,g}} \right)$$
$$\text{subject to: } U + \sum_{g \in \mathcal{P}(\mathcal{S})} (L_g + b_g) \leq T$$
$$b_g \geq 0, \ \forall g \in \mathcal{P}(\mathcal{S})$$
$$\tag{14}$$

Considering the inner minimization problem for a given $\mathcal{P}(\mathcal{S})$, we can rewrite it as follows:

$$\min_{\mathbf{b}} \ln \Gamma(\mathcal{P}(\mathcal{S}), \mathbf{b}) + \sum_{g \in \mathcal{P}(\mathcal{S})} \ln \left( F_{P,g} + (1 - F_{P,g})\alpha^{b_g/F_{N,g}} \right)$$
$$\text{s. t.: } U + \sum_{g \in \mathcal{P}(\mathcal{S})} (L_g + b_g) \leq T$$
$$b_g \geq 0, \ \forall g \in \mathcal{P}(\mathcal{S}).$$
$$\tag{15}$$

By introducing a multiplier $\lambda \leq 0$, we can write the augmented Lagrangian as

$$\mathcal{L}(\mathbf{b}, \lambda) = \ln \Gamma(\mathcal{P}(\mathcal{S}), \mathbf{b}) + \sum_{g \in \mathcal{P}(\mathcal{S})} \ln \left( F_{P,g} \right.$$
$$\left. + (1 - F_{P,g})\alpha^{b_g/F_{N,g}} \right) - \lambda \left( \sum_{g \in \mathcal{P}(\mathcal{S})} (L_g + b_g) - T + U \right).$$
$$\tag{16}$$

Denote $\phi_g \triangleq \partial \ln \Gamma(\mathcal{P}(\mathcal{S}), \mathbf{b})/\partial b_g$. The partial derivatives of $\mathcal{L}$ w.r.t. $b_g$ and $\lambda$ can be derived as

$$\begin{cases} \dfrac{\partial \mathcal{L}}{\partial b_g} = \phi_g + \dfrac{(1 - F_{P,g})\alpha^{b_g/F_{N,g}} \ln \alpha}{F_{N,g}\left(F_{P,g} + (1 - F_{P,g})\alpha^{b_g/F_{N,g}}\right)} - \lambda \\ \dfrac{\partial \mathcal{L}}{\partial \lambda} = T - U - \sum_{g \in \mathcal{P}(\mathcal{S})} (L_g + b_g) \end{cases}$$
$$\tag{17}$$

By imposing $\partial \mathcal{L}/\partial b_g = 0$, we obtain

$$\frac{(1 - F_{P,g})\alpha^{b_g/F_{N,g}}}{F_{N,i}\left(F_{P,g} + (1 - F_{P,g})\alpha^{b_g/F_{N,g}}\right)} = \frac{\lambda}{\ln \alpha} - \frac{\phi_g}{\ln \alpha}$$
$$\triangleq c - \Phi_g, \ \forall g \in \mathcal{P}(\mathcal{S}). \tag{18}$$

In the above equation, we define an auxiliary variable $c$ as in the analysis of the previous section, and denote $\Phi_g \triangleq$

$\phi_g/\ln \alpha$. We can derive the optimal $b_g^*$ as

$$b_g^* = F_{N,g} \log_\alpha \frac{(c - \Phi_g^*)F_{N,g}F_{P,g}}{(1 - (c - \Phi_g^*)F_{N,g})(1 - F_{P,g})}. \tag{19}$$

Here, $b_g^*$ can be solved iteratively. Specifically, we define an auxiliary variable

$$u_g(t) \triangleq b_g(t) -$$
$$F_{N,g} \log_\alpha \frac{(c - \Phi_g(t))F_{N,g}F_{P,g}}{\left(1 - (c - \Phi_g(t))F_{N,g}\right)(1 - F_{P,g})},$$

where $t$ denotes the corresponding iteration index. We employ the secant method (Barnes, 1965) to update $b_g(t)$ as follows:

$$b_g(t+1) = b_g(t) - \frac{u_g(t)}{u_g(t) - u_g(t-1)} (b_g(t) - b_g(t-1)).$$

Noticing that the derivative of $u_g(t)$ is non-zero in the vicinity of $b_g^*$, and assuming that $\Gamma(\mathcal{P}(\mathcal{S}), \mathbf{b})$ is continuous and differentiable, we can prove that $b(t)$ converges to $b_g^*$, armed with which we can derive the overall FPR, denoted by $F^{(\mathcal{S})}$ as follows:

$$F^{(\mathcal{S})} = \Gamma(\mathcal{P}(\mathcal{S}), \mathbf{b}^*) \prod_{g \in \mathcal{P}(\mathcal{S})} F_P^{(g)}$$
$$= \Gamma(\mathcal{P}(\mathcal{S}), \mathbf{b}^*) \prod_{g \in \mathcal{P}(\mathcal{S})} \frac{F_{P,g}}{1 - (c - \Phi_g^*)F_{N,g}}. \tag{20}$$

By searching $c$ in $(0, \min_{g \in \mathcal{P}(\mathcal{S})}(1 - F_{P,g})/F_{N,g}]$ and finding the minimal corresponding overall FPR, we can solve the inner optimization problem. The detailed pseudo-code is given in Algorithm 3 of Appendix D.

Next, we turn to solve the outer optimization problem by developing a greedy merging algorithm, described as follows:

- **Initialization**. We set $\mathcal{P}$ as an empty set.
- **Augmenting $\mathcal{P}$**. We iteratively augment $\mathcal{P}$ using the following three strategies:
  - **Creating new branch**: We pick an oracle not yet selected and create a new branch.
  - **Augmenting existing branch**: We pick an oracle not yet selected and merge it with one of existing branches in $\mathcal{P}$.
  - **Merging two existing branches**: We pick two branches in $\mathcal{P}$ and merge them.

For each of the above augmentation strategy, we invoke our algorithm to solve the inner optimization problem under the augmented partition $\mathcal{P}$. We stop the augmentation when the overall FPR cannot be further improved. At this point, we output the final partition and the corresponding backup filter size to set the ELBF++ parameters.

We conclude this section by analyzing the time complexity of the greedy algorithm. Given the error bound $\epsilon_b$ on finding $b_g^*$, the number of iterations to execute the secant-based update is $O\left(\log_\xi\left(\log\left(1/\epsilon_b\right)\right)\right)$, where $\xi = (1+\sqrt{5})/2$ (Díez, 2003). Hence, for a given $\mathcal{P}$, the complexity of finding the minimal overall FPR among all discretized values of $c$ is $O\left(\left(\log_\xi\left(\log\left(1/\epsilon_b\right)\right)\right)/\Delta\right)$. Given that the inner optimization procedure is invoked for each augmentation of the partition $\mathcal{P}$, and considering that the augmentation steps may be executed up to $n_0$ times, the overall time complexity sums up to $O\left(n_0^3\left(\log_\xi\left(\log\left(1/\epsilon_b\right)\right)\right)/\Delta\right)$.

## 5. Extensions and Variants

In this section, we discuss several pertinent extensions and variants of our ELBF to demonstrate that our design is sufficiently generic to be applied in a large spectrum of data processing tasks: (1) We formulate and optimize sandwiched ELBF which is incorporated with initial filters prior to oracles; (2) We consider an enhanced scenario where each oracle can be optimized as well; (3) We propose Ensemble Learned Bloomier Filter, which is inspired by Bloomier filters (Chazelle et al., 2004), a variant of BF, to solve key-value lookup. The details is presented in the Appendix.

## 6. Experiments

We perform empirical experiments to evaluate the proposed designs. We conduct our experiments on three practical data analysis tasks: (1) virus signature scan, (2) malicious URLs detection, (3) universally unique identifier check. Due to space limit, we present the first task and defer the rest two to Appendix, which demonstrate similar results and thus lead to similar observations and conclusions. We also give a more detailed evaluation of Algorithm 1 under different parameters using synthetic data. A aberration study presented in Appendix compares Algorithm 1 against simpler approaches, highlighting the advantage of our proposition. Our experiments are conducted on a standard off-the-shelf desktop computer with an Intel(R) Core i7-12700F CPU @2.10 GHz and 16 GB RAM.

### 6.1. Experiment Setting

BFs are extensively utilized in virus signature scanning task, involving the verification of whether a suspect file's signature exists in the virus signature database when the file is presented. In our experiments, we apply the proposed ELBFs to execute the task of virus signature scan and compare them against the state-of-the-art solutions.

We use a real-world dataset EMBER (Anderson & Roth, 2018), an open-source collection of 1M sha256 file hashes. There are 400K malicious, 400K benign and 200K unlabeled files, which were scanned by VirusTotal in 2018. We

ignore the unlabeled files. The 2381 features of the files are included in the dataset, from which we randomly select 238 features. The features are exclusive for mutually independent case and somewhat duplicate for correlated case. Following the literature, we choose the random forest classifier from sklearn (Pedregosa et al., 2011) as oracles. We construct a pool of oracles with a large oracle and a set of small oracles. The large oracle consists 10 decision trees and at most 20 leaf nodes for each tree; the oracle size is 262Kb. Each small oracle is a single decision tree with at most 20 leaf nodes; the oracle size is 32Kb. We randomly sample 30% data for oracle training. The large oracle learns from all 238 features while each small oracle learns from 23 or 24 features selected randomly.

### 6.2. Experiment Results

We compare ELBFs against major state-of-the-art propositions, i.e., the canonical BF (Bloom, 1970), LBF (Kraska et al., 2018), Ada-BF (Dai & Shrivastava, 2020), PLBF (Vaidya et al., 2021) and Fast PLBF++ (Sato & Matsui, 2023). As we have a pool of oracles, we let LBF, Ada-BF, PLBF and Fast PLBF++ operate on the oracle achieving the lowest FPR. To put Ada-BF, PLBF and Fast PLBF++ along the same line of comparison with ELBF, we let Ada-BF, PLBF and Fast PLBF++ operate on two regions representing positive and negative responses from the oracle and optimize the corresponding backup BFs. Ada-BF executes under its hyper-parameter $c = 1 - 3$ with 40 intervals and PLBF and Fast PLBF++ execute under its number of intervals $N = 100$. ELBF chooses the oracles by running Algorithm 1 under the parameter setting $\Delta = 0.1$, $\epsilon = 0.01$, $\delta = 0.01$. We straightforwardly set theta $\theta_1 = \left\lceil \frac{n_o}{\epsilon} \right\rceil$ and $\theta_2 = \left\lceil \frac{n_o+1}{T\delta} \right\rceil$, which satisfies the condition in Theorem 3.2. ELBF++ operates the greedy merging algorithm under the parameter setting $\Delta = 0.1$, $\epsilon_b = 0.01$.

#### 6.2.1. MEMORY BUDGET AND FPR

We first compare the central performance metric, FPR, of the evaluated solutions in Figure 4(a) and Figure 4(c) under different memory budget. We make the following observations from the results. (1) Under stringent memory budget, the canonical BF has much higher FPR compared to its learning-augmented peers. This performance gap shrinks with the increase of the memory budget to the extent that, with abundant memory space, there is no advantage to augment the BF with oracles. This can be explained as follows. The FPR of the canonical BF decreases exponentially in the memory allocated to it. In the LBF, Ada-LBF, PLBF and Fast PLBF++, the FPR depends not only on the size of the backup BF, but also on the FPR of the oracles. When the memory budget is not stringent, the FPR of the oracles dominates the global FPR, leading to the result that the canonical BF outperforms its learning-augmented peers, except our solution ELBFs,

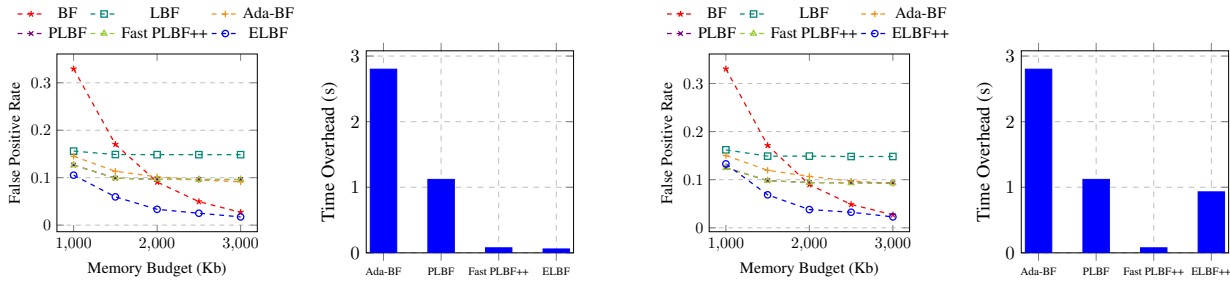

*Figure 4.* Performance comparison: mutually independent case (left), correlated case (right).

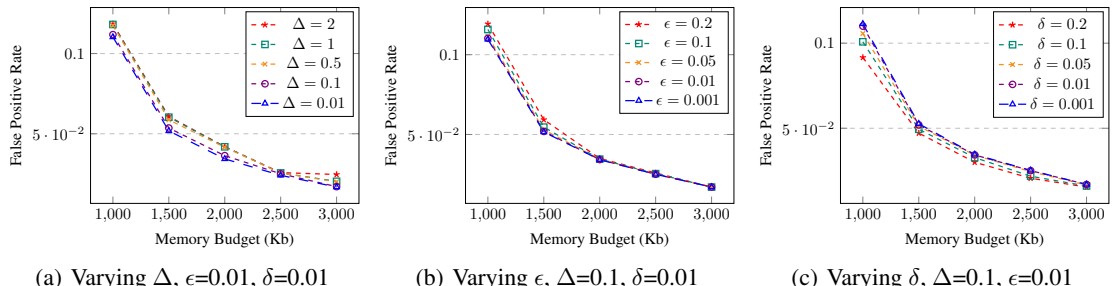

| (a) Varying $\Delta$, $\epsilon$=0.01, $\delta$=0.01 | (b) Varying $\epsilon$, $\Delta$=0.1, $\delta$=0.01 | (c) Varying $\delta$, $\Delta$=0.1, $\epsilon$=0.01 |
|---|---|---|

*Figure 5.* Performance of Algorithm 1 under different parameter settings.

as integrating multiple different oracles eliminate the diminishing effect of a single oracle and we have included a dumb oracle allowing to cover the case where adding an oracle has negative impact on the final FPR; (2) ELBFs outperform the other solutions for all the memory budget, confirming our finding that orchestrating multiple small oracles instead of relying on a single large oracle holistically can effectively reduce the FPR. The performance gain is more pronounced with large memory budget because this allows more space to optimize the orchestration of different oracles, leading to better performance. (3) The performance of Ada-BF, PLBF, and Fast PLBF++ is unrelated to the correlation of oracles, as they only use a single oracle. ELBF performs better than ELBF++ in their respective situations, as ELBF achieves guaranteed optimality while ELBF++ relies on heuristics.

### 6.2.2. TIME OVERHEAD

We then compare the time overhead of different solutions, by time overhead we mean the time to compute the optimal configuration of the entire data structure including the backup filter size, the oracles to use, etc, excluding the times to train models and insert all the items into the BF and backup BFs. In our experiments, we report that the time overhead for BF and LBF is negligible and hence have not included them in the figure. This is because BF does not need to choose any oracle and LBF chooses the best oracle with minimal overall FPR, both with almost no computation overhead. Therefore, we concentrate on the comparison among ELBFs, Ada-BF, PLBF and Fast PLBF++, as depicted in Figure 4(b) and Figure 4(d). We observe that both Ada-BF and PLBF incur significantly higher overhead than

ELBF, because both of them require multiple-dimensional search for the parameters, outweighing the Knapsack-based search in ELBF. Though Fast PLBF++ diminishes the construction time of PLBF, its time overhead remains higher than that of ELBF. ELBF++ is more time-consuming than ELBF due to the heuristics.

### 6.2.3. ABLATION STUDY W.R.T. ALGORITHM 1

We conclude our experiment analysis by evaluating Algorithm 1 under different parameter settings regarding $c$, $\epsilon$ and $\delta$. The results, depicted in Figure 5, lead to the following observations. The FPR increases in $\Delta$ and $\epsilon$, demonstrating the trade-off between the optimality and complexity of our algorithm. Our experiments indicate that setting $\Delta$ to 0.1 seems sufficient to achieve near-optimal FPR. On the other hand, the FPR decreases in $\delta$, as larger $\delta$ indicates larger memory budget and hence better performance. The impact of $\delta$ decreases as the memory budget increases, to the extent that, when the memory budget reaches 3000 Kb, $\delta$ has virtually no impact on the FPR.

## 7. Conclusion

In this paper, we have presented a new design of LBFs based on ensemble learning, termed as ELBFs. We demonstrate that by orchestrating multiple oracles of smaller size, the ELBFs outperform the existing LBFs. The main technicality to build the optimal ELBFs in our designs is a Knapsack-based algorithm solving a combinatorial optimization problem for mutually independent learning oracles and a greedy approach solving a more complex combinatorial optimiza-

tion problem for correlated learning oracles. The empirical experiments in real-world datasets demonstrate the performance gain of our approaches.

## Acknowledgements

This work is supported in part by National Science Foundation of China under Grant 62172455, Pearl River Talent Program under Grant 2019QN01X140, and Guangdong Provincial Key Laboratory of Information Security Technology (No. 2023B1212060026). Part of the work of Lin Chen is done when he was with School of Computer Science and Engineering, Sun Yat-sen University.

## Impact Statement

This paper presents work whose goal is to advance the field of Machine Learning. There are a number of potential societal consequences of our work, none which we feel must be specifically highlighted here.

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

# A. Proof of Theorem 3.2

*Proof.* Let $(\mathcal{A}^*, \mathbf{b}^*)$ and $(\mathcal{A}^{OPT}, \mathbf{b^{OPT}})$ denote the output of Algorithm 1 and the optimal solution of $\mathbf{P_1}$, respectively. Let $c^*$ and $c^{OPT}$ denote the associated values of $c$ correspondingly.

We consider the discretized Knapsack formulation of $\mathbf{P}$ denoted by $\hat{\mathbf{P}}$, in which $\hat{\mathbf{v}}$ and $\hat{\mathbf{w}}$ are computed using scaling factors $\theta_1$ and $\theta_2$ as shown in Algorithm 1. Recall the scaling of $w_j$, we can check that $(\mathcal{A}^{OPT}, \mathbf{b}')$ is a feasible solution of $\hat{\mathbf{P}}$, where $\mathbf{b}'$ is as follows

$$b_i' = F_{N,i} \log_\alpha \frac{\hat{c}^{OPT} F_{N,i} F_{P,i}}{(1 - \hat{c}^{OPT} F_{N,i})(1 - F_{P,i})}$$

$$\geq F_{N,i} \log_\alpha \frac{c^{OPT} F_{N,i} F_{P,i}}{(1 - c^{OPT} F_{N,i})(1 - F_{P,i})},$$

where $\hat{c}^{OPT} = \left\lfloor \frac{c^{OPT}}{\Delta} \right\rfloor \cdot \Delta \leq c^{OPT}$.

Hence, we have

$$\sum_{i \in \mathcal{A}^*} \left\lceil \theta_1 \cdot \ln \frac{1 - c^* F_{N,i}}{F_{P,i}} \right\rceil \geq \sum_{i \in \mathcal{A}^*} \left\lceil \theta_1 \cdot \ln \frac{1 - \hat{c}^{OPT} F_{N,i}}{F_{P,i}} \right\rceil$$

$$\geq \sum_{i \in \mathcal{A}^*} \left\lceil \theta_1 \cdot \ln \frac{1 - c^{OPT} F_{N,i}}{F_{P,i}} \right\rceil.$$

The second inequality follows from $\hat{c}^{OPT} \leq c^{OPT}$. It then follows that

$$\sum_{i \in \mathcal{A}^*} \left( 1 + \theta_1 \cdot \ln \frac{1 - c^* F_{N,i}}{F_{P,i}} \right) \geq \sum_{i \in \mathcal{A}^*} \theta_1 \cdot \ln \frac{1 - c^{OPT} F_{N,i}}{F_{P,i}}.$$

Let $n_0$ denote the number of oracles. We have

$$\sum_{i \in \mathcal{A}^*} \ln \frac{1 - c^* F_{N,i}}{F_{P,i}} \geq \sum_{i \in \mathcal{A}^*} \ln \frac{1 - c^{OPT} F_{N,i}}{F_{P,i}} - \frac{n_0}{\theta_1}.$$

Recall that the false positive rates corresponding to $(\mathcal{A}^*, \mathbf{b}^*)$ and $(\mathcal{A}^{OPT}, \mathbf{b^{OPT}})$ are $\prod_{i \in \mathcal{A}^*} \frac{F_{P,i}}{1 - c^* F_{N,i}}$ and $\prod_{i \in \mathcal{A}^*} \frac{F_{P,i}}{1 - c^{OPT} F_{N,i}}$, respectively. With some straightforward algebraic operations, we can prove that, under the condition $\theta_1 \geq n_0/\epsilon$, it holds that

$$\prod_{i \in \mathcal{A}^*} \frac{F_{P,i}}{1 - c^* F_{N,i}} \leq (1 + \epsilon) \prod_{i \in \mathcal{A}^*} \frac{F_{P,i}}{1 - c^{OPT} F_{N,i}}.$$

We have now proved the false positive rate of the solution output by our algorithm is within $\epsilon$ to the optimal solution. We then prove that the total space of our solution is upper-bounded by $(1 + \delta)T$.

It follows from our algorithm that

$$\sum_{i \in \mathcal{A}^*} \lfloor \theta_2 (L_i + b_i^*) \rfloor \leq \lceil \theta_2 T \rceil.$$

Therefore, we have

$$\sum_{i \in \mathcal{A}^*} \theta_2 (L_i + b_i^*) - n_0 \leq \theta_2 T + 1.$$

The space overhead of our algorithm can then be bounded as follows.

$$\sum_{i \in \mathcal{A}^*} (L_i + b_i^*) \leq T + \frac{n_0 + 1}{\theta_2}.$$

Therefore, if $\theta_2 \geq \frac{n_0 + 1}{T\delta}$, the total space of our solution is upper-bounded by $(1 + \delta)T$. $\qquad \square$

## B. Optimality of ELBF with Non-correlated Oracles

**Theorem B.1.** *Given any set of independent oracles and any space budget, allocating a separate backup filter for each oracle leads to strictly lower FPR than letting multiple oracles sharing a common backup filter.*

We consider the oracles are divided into a set $\mathcal{G}$ of $n_g \triangleq |\mathcal{G}|$ groups, with the oracles belonging to the same group sharing a single backup filter, i.e., the filter of group $g \in \mathcal{G}$ stores the false negatives of all the oracles in this group.

We now derive the FPR by first deriving the FPR of each group and then deriving the overall FPR. For each group $g \in \mathcal{G}$, once at least one of the oracles in the group returns negatively, the backup filter will store the corresponding item. Hence, the FNR of group $g$ oracles as a whole, denoted by $F_N^{(g)}$, can be computed as

$$F_N^{(g)} = 1 - \prod_{i \in \mathcal{O}_g} (1 - F_{N,i}), \tag{21}$$

where $\mathcal{O}_g$ denotes the set of oracles in group $g$. The number of keys the backup filter of group $g$ needs to hold is $F_N^{(g)} n$. To derive the FPR for group $g$, we first give the FPR for the oracles in the group, denoted by $F_P^{(g)}$ as follows.

$$F_P^{(g)} = \prod_{i \in \mathcal{O}_g} F_{P,i} \tag{22}$$

We then can derive the FPR for the whole branch including the backup filter as

$$F_P^{(g)} + (1 - F_P^{(g)}) \alpha^{b^{(g)}/F_N^{(g)}}.$$

Since the groups are independent to each other, we can conclude the FPR for this group-based design as

$$\prod_{g \in \mathcal{G}} \left( F_P^{(g)} + (1 - F_P^{(g)}) \alpha^{b^{(g)}/F_N^{(g)}} \right) \tag{23}$$

We next prove that for any configuration of this group-based design, there always exists a corresponding ELBF configuration achieving lower FPR with the same space budget.

Suppose we have in total $n$ oracles, with $F_{P,i}$ and $F_{N,i}$ denoting the FPR and the FNR of oracle $i$ and $L_i$ denoting its size. The space budget is $T$. We start from the case $n = 2$. The FPR of the group containing both oracles can be computed as

$$F_{\mathcal{G}} = F_{P,1} F_{P,2} + (1 - F_{P,1} F_{P,2}) \alpha^{b/(F_{N,1} + F_{N,2} - F_{N,1} F_{N,2})},$$

where $b \triangleq T - L_1 - L_2$. The FPR of the corresponding ELBF is $F_1 \cdot F_2$ that given by

$$\left( F_{P,1} + (1 - F_{P,1}) \alpha^{b_1/F_{N,1}} \right) \left( F_{P,2} + (1 - F_{P,2}) \alpha^{b_2/F_{N,2}} \right),$$

where $b_1 + b_2 = b$.

Now, we need to prove $F_1 \cdot F_2 < F_{\mathcal{G}}$. Imposing $\frac{b_1}{F_{N,1}} = \frac{b_2}{F_{N,2}}$ yields $b_1 = \frac{F_{N,1} b}{F_{N,1} + F_{N,2}}, b_2 = \frac{F_{N,2} b}{F_{N,1} + F_{N,2}}$. The inequality can be rewrite as below.

$$\left( F_{P,1} + (1 - F_{P,1}) \alpha^{\frac{b}{F_{N,1} + F_{N,2}}} \right) \cdot \left( F_{P,2} + (1 - F_{P,2}) \alpha^{\frac{b}{F_{N,1} + F_{N,2}}} \right) < F_{P,1} F_{P,2} + (1 - F_{P,1} F_{P,2}) \alpha^{\frac{b}{F_{N,1} + F_{N,2} - F_{N,1} F_{N,2}}}$$

$$\Longleftrightarrow (F_{P,1} + F_{P,2} - 2F_{P,1} F_{P,2}) \alpha^{\frac{b}{F_{N,1} + F_{N,2}}} + (1 - F_{P,1})(1 - F_{P,2}) \alpha^{\frac{2b}{F_{N,1} + F_{N,2}}} < (1 - F_{P,1} F_{P,2}) \alpha^{\frac{b}{F_{N,1} + F_{N,2} - F_{N,1} F_{N,2}}}$$

$$\Longleftrightarrow (F_{P,1} + F_{P,2} - 2F_{P,1} F_{P,2}) \alpha^{1 - \frac{F_{N,1} F_{N,2}}{F_{N,1} + F_{N,2}}} + (1 - F_{P,1})(1 - F_{P,2}) \alpha^{2(1 - \frac{F_{N,1} F_{N,2}}{F_{N,1} + F_{N,2}})} < 1 - F_{P,1} F_{P,2}.$$

Since $1 - \frac{F_{N,1} F_{N,2}}{F_{N,1} + F_{N,2}} \in (0, 1)$, we have $\alpha^{1 - \frac{F_{N,1} F_{N,2}}{F_{N,1} + F_{N,2}}} < 1$. The LHS of the above inequality is strictly smaller than

$$F_{P,1} + F_{P,2} - 2F_{P,1} F_{P,2} + (1 - F_{P,1})(1 - F_{P,2}) = 1 - F_{P,1} F_{P,2}.$$

The inequality is thus proven. We carry on the proof by mathematical induction.

Suppose $n = k$, $\exists b_i$, where $\sum_{i=1}^{k} b_i = T - \sum_{i=1}^{k} L_i$, s.t. $\prod_{i=1}^{k} F_i < F_{\mathcal{G}_k}$.

For $n = k + 1$, the false positive rate of group $G_{k+1}$ is

$$F_{P,\mathcal{G}_{k+1}} = \prod_{i=1}^{k+1} F_{P,i} = \prod_{i=1}^{k} F_{P,i} \cdot F_{P,k+1} = F_{P,\mathcal{G}_k} \cdot F_{P,k+1} \tag{24}$$

and the false negative rate of that is

$$
\begin{aligned}
F_{N,\mathcal{G}_{k+1}} &= 1 - \prod_{i=1}^{k+1}(1 - F_{N,i}) \\
&= 1 - \left(1 - \left(1 - \prod_{i=1}^{k}(1 - F_{N,i})\right)\right) \cdot (1 - F_{N,k+1}) \\
&= 1 - (1 - F_{N,\mathcal{G}_k})(1 - F_{N,k+1}) \\
&= F_{N,\mathcal{G}_k} + F_{N,k+1} - F_{N,\mathcal{G}_k} F_{N,k+1}.
\end{aligned}
\tag{25}
$$

from which we can write the FPR of groups consisting $k + 1$ oracles as

$$F_{P,\mathcal{G}_k} F_{P,k+1} + (1 - F_{P,\mathcal{G}_k} F_{P,k+1}) \alpha^{\frac{T}{F_{N,\mathcal{G}_k} + F_{N,k+1} - F_{N,\mathcal{G}_k} F_{N,k+1}}} \tag{26}$$

That is, the ahead $k$ oracles that compose the group can be treated as an alternative oracle. We need to prove there exists an allocation of BFs for multiple oracles which behaves better than the group case.

Recall $b = T - \sum_{i=1}^{k+1} L_i$. Imposing $\frac{b_{\mathcal{G}_k}}{F_{N,\mathcal{G}_k}} = \frac{b_{k+1}}{F_{N,k+1}}$ yields $b_{\mathcal{G}_k} = \frac{F_{N,\mathcal{G}_k} b}{F_{N,\mathcal{G}_k} + F_{N,k+1}}$, $b_{k+1} = \frac{F_{N,k+1} b}{F_{N,\mathcal{G}_k} + F_{N,k+1}}$. Taking the assumption for $n = k$, we know that there exists an allocation for BFs for the ahead $k$ oracles under budget $b_{\mathcal{G}_k}$ satisfies the following inequality.

$$
\begin{aligned}
\prod_{i=1}^{k+1} F_i &= \prod_{i=1}^{k} F_i \cdot F_{k+1} < F_{\mathcal{G}_k} \cdot F_{k+1} \\
&= \left(F_{P,\mathcal{G}_k} + (1 - F_{P,\mathcal{G}_k}) \alpha^{b_{\mathcal{G}_k}/F_{N,\mathcal{G}_k}}\right) \cdot \left(F_{P,k+1} + (1 - F_{P,k+1}) \alpha^{b_{k+1}/F_{N,k+1}}\right) \\
&= \left(F_{P,\mathcal{G}_k} + (1 - F_{P,\mathcal{G}_k}) \alpha^{\frac{b}{F_{N,\mathcal{G}_k} + F_{N,k+1}}}\right) \cdot \left(F_{P,k+1} + (1 - F_{P,k+1}) \alpha^{\frac{b}{F_{N,\mathcal{G}_k} + F_{N,k+1}}}\right)
\end{aligned}
\tag{27}
$$

This inequality is smaller than (26) by similar analysis for the case $n = 2$. Then, we have $F_{\mathcal{G}_k} \cdot F_{k+1} < F_{\mathcal{G}_{k+1}}$, which leads to

$$\prod_{i=1}^{k+1} F_i < F_{\mathcal{G}_{k+1}}$$

Thus, given mutually independent oracles, for any configuration of the group-based design, there always exists a corresponding ELBF configuration achieving lower FPR with the same space budget.

## C. Algorithm 2: Training Auxiliary Learned Model

## D. Algorithm 3: Inner Optimization of ELBF++

## E. Algorithm 4: Greedy Merging Algorithm for Outer Optimization of ELBF++

The pseudo-code of our greedy merging algorithm for the outer optimization of ELBF++ is given in Algorithm 4. Our three augmentation strategies correspond to three **for** loops, respectively. The procedure INNER defined in Algorithm 3 solves the inner optimization problem. For conciseness, an internal procedure UPDATE updates the current optimal configuration.

---

**Algorithm 2** Training auxiliary learned model

---

**Input:** oracles set $\mathcal{O}$, space budget $T$, data scaler $M$
**Output:** auxiliary model $\Gamma$

1: **Initialization:** $i \leftarrow 1$
2: **while** $1 \leq i \leq M$ **do**
3:    $j \leftarrow 1$
4:    **while** $1 \leq j \leq n_0$ **do**
5:      $s \leftarrow randomInt(j, n_0)$
6:      Randomly divide $\mathcal{O}$ into a partition $\mathcal{P}(\mathcal{O})$ containing $s$ groups
7:      $\mathcal{P}(\mathcal{S}) \leftarrow$ Randomly select a subset containing $j$ groups of $\mathcal{P}(\mathcal{O})$
8:      $\mathbf{b} \triangleq (b_1, b_2, \cdots, b_{n_0}) \leftarrow \mathbf{0}$
9:      Randomly allocate $T$ to $b_1, b_2, \cdots, b_j$
10:     $R \leftarrow \dfrac{\Pr(X_1, X_2, \cdots, X_j)}{\prod_{i=1}^{j} \Pr(X_i)}$
11:      $\mathbf{x} \leftarrow$ Extract features from $\mathcal{P}(\mathcal{S})$
12:     Add $((\mathbf{x}, \mathbf{b}), R)$ to dataset
13:    **end while**
14: **end while**
15: $\Gamma \leftarrow$ Train a Random Forest on dataset

---

**Algorithm 3** Inner Optimization of ELBF++

---

1: **procedure** INNER(oracle partition $\mathcal{P}$, stepsize $\Delta$, space budget $T$, auxiliary model $\Gamma$, iteration threshold $N$)
2:    $F_P^{(\mathcal{P})} \leftarrow 1, c \leftarrow 0, U \leftarrow$ size of $\Gamma$
3:    **while** $0 \leq c \leq \min\limits_{g \in \mathcal{P}(\mathcal{S})} \dfrac{1 - F_{P,g}}{F_{N,g}}$ **do**
4:      $c \leftarrow c + \Delta$
5:      **for** $g \in \mathcal{P}$ **do**
6:         Initialize $\mathbf{b}$ randomly
7:         Compute $\Phi_g(0), \forall g \in \mathcal{P}$
8:         $t \leftarrow 0$
9:         **while** convergence not reached and $t < N$ **do**
10:           $u_g(t) \leftarrow b_g(t) - F_{N,g} \log_\alpha \dfrac{(c - \Phi_g(t)) F_{N,g} F_{P,g}}{(1 - (c - \Phi_g(t)) F_{N,g})(1 - F_{P,g})}$
11:           **if** $t > 0$ **then**
12:              $b_g(t+1) \leftarrow b_g(t) - \dfrac{u_g(t)}{u_g(t) - u_g(t-1)}(b_g(t) - b_g(t-1))$
13:           **end if**
14:           $t \leftarrow t + 1$
15:         **end while**
16:         $b_g^* \leftarrow b_g(t),\ \Phi_g^* \leftarrow \Phi_g(t)$
17:      **end for**
18:      $F_P^{(\mathcal{P})} \leftarrow \Gamma(\mathcal{P}, \mathbf{b}^*) \prod\limits_{g \in \mathcal{P}} \dfrac{F_{P,g}}{1 - (c - \Phi_g^*)F_{N,g}},\ w \leftarrow \sum\limits_{g \in \mathcal{P}} b_g^*$
19:      **if** $F_P^{(\mathcal{P})} < F_P^{(\mathcal{P})*}$ and $w \leq T - U$ **then**
20:         $F_P^{(\mathcal{P})*} \leftarrow F_P^{(\mathcal{P})}$
21:      **end if**
22:    **end while**
23:    **return** $\mathbf{b}^*, F_P^{(\mathcal{P})*}$
24: **end procedure**

---

---

**Algorithm 4** Greedy Merging Algorithm for Outer Optimization of ELBF++

---

**Require:** Oracles set $\mathcal{O}$, space budget $T$, stepsize $\Delta$, iteration threshold $N$
**Ensure:** Optimal partition $\mathcal{P}^*$ and corresponding $\mathbf{b}^*$
1: **Initialization:** $F_P^* \leftarrow 1, \mathcal{P} \leftarrow \emptyset$
2: **procedure** UPDATE$(\mathcal{P}_{\text{new}}, \mathbf{b}_{\text{new}}^*, F_{P,\text{new}}^*)$
3:     $\mathcal{P}^* \leftarrow \mathcal{P}_{\text{new}}$
4:     $\mathbf{b}^* \leftarrow \mathbf{b}_{\text{new}}^*$
5:     $F_P^* \leftarrow F_{P,\text{new}}^*$
6:     **return** True, $\mathcal{P}^*$, $\mathbf{b}^*$, $F_P^*$
7: **end procedure**
8: **while** True **do**
9:     $imp \leftarrow$ False
10:     **for** each oracle $o$ not in $\mathcal{P}$ **do**
11:         $\mathcal{P}_{\text{new}} \leftarrow \mathcal{P} \cup \{o\}$
12:         $\mathbf{b}_{\text{new}}^*, F_{P,\text{new}}^* \leftarrow$ INNER$(\mathcal{P}_{\text{new}}, \Delta, T, \Gamma, N)$
13:         **if** $F_{P,\text{new}}^* < F_P^*$ **then**
14:             $(imp, \mathcal{P}^*, \mathbf{b}^*, F_P^*) \leftarrow$ UPDATE$(\mathcal{P}_{\text{new}}, \mathbf{b}_{\text{new}}^*, F_{P,\text{new}}^*)$
15:         **end if**
16:     **end for**
17:     **for** each branch $g \in \mathcal{P}$ **do**
18:         **for** each oracle $o$ not in $\mathcal{P}$ **do**
19:             $\mathcal{P}_{\text{new}} \leftarrow \mathcal{P} \setminus \{g\} \cup \{g \cup \{o\}\}$
20:             $(\mathbf{b}_{\text{new}}^*, F_{P,\text{new}}^*) \leftarrow$ INNER$(\mathcal{P}_{\text{new}}, \Delta, T, \Gamma, N)$
21:             **if** $F_{P,\text{new}}^* < F_P^*$ **then**
22:                 $(imp, \mathcal{P}^*, \mathbf{b}^*, F_P^*) \leftarrow$ UPDATE$(\mathcal{P}_{\text{new}}, \mathbf{b}_{\text{new}}^*, F_{P,\text{new}}^*)$
23:             **end if**
24:         **end for**
25:     **end for**
26:     **for** each pair of branches $(g_1, g_2) \in \mathcal{P}$ **do**
27:         $\mathcal{P}_{\text{new}} \leftarrow \mathcal{P} \setminus \{g_1, g_2\} \cup \{g_1 \cup g_2\}$
28:         $(\mathbf{b}_{\text{new}}^*, F_{P,\text{new}}^*) \leftarrow$ INNER$(\mathcal{P}_{\text{new}}, \Delta, T, \Gamma, N)$
29:         **if** $F_{P,\text{new}}^* < F_P^*$ **then**
30:             $(imp, \mathcal{P}^*, \mathbf{b}^*, F_P^*) \leftarrow$ UPDATE$(\mathcal{P}_{\text{new}}, \mathbf{b}_{\text{new}}^*, F_{P,\text{new}}^*)$
31:         **end if**
32:     **end for**
33:     **if** not $imp$ **then**
34:         Break
35:     **end if**
36:     $\mathcal{P} \leftarrow \mathcal{P}^*$
37: **end while**
38: **return** $\mathcal{P}^*$, $\mathbf{b}^*$, $F_P^*$

---

# F. Details of Extensions and Variants

## F.1. Optimizing Learned Oracles

Referring to Problem 1, we consider an enhanced scenario where each oracle possesses variable size and performance characteristics. For instance, a decision tree oracle might incorporate additional leaf nodes to reduce its false positives and false negatives, albeit at the expense of increased memory consumption. In this context, the objective is to select an optimal configuration for each oracle such that the resultant ELBF achieves the minimal FPR.

Assume that each oracle $i \in \mathcal{O}$ has a set $\mathcal{J}_i$ of potential configurations. Each configuration $j \in \mathcal{J}_i$ is associated with specific parameters: $L_{ij}$ denotes the memory size, $F_{P,ij}$ represents the false positive rate, and $F_{N,ij}$ represents the false negative rate of the oracle under configuration $j$. The challenge is to determine the most appropriate configuration for each oracle to optimize the overall FPR of the ELBF.

Problem 1 has evolved in complexity, now encompassing the optimization of individual configurations for each oracle $i$ within the chosen subset $\mathcal{A}$. Reflecting on Algorithm 1, it is capable of addressing this intricate optimization challenge. Given that the values of $F_{P,i}$ and $F_{N,i}$ fluctuate with the configuration $j \in \mathcal{J}_i$ of oracle $i$, the variables $b_i^*$ in (9) and $F_P^{(i)}$ in (10) are redefined as $b_{ij}^*$ and $F_P^{(ij)}$, respectively.

In the context of the Knapsack Problem formulation (11), each oracle's value now ranges over the set $\{\ln \frac{1-cF_{N,ij}}{F_{P,ij}} | j \in \mathcal{J}_i\}$, and its weight is recalculated over the set $\{L_{ij} + b_{ij}^* | j \in \mathcal{J}_i\}$. This reformulation suggests that the problem can be cast as a Grouped Knapsack Problem, a derivative of the classic Knapsack Problem, solvable in pseudo-polynomial time via dynamic programming. In this framework, each oracle is no longer a single object but a group of objects, with each object corresponding to a specific configuration. We aim to select at most one object from each group to maximize the value of the objective function.

Within the **for** loop of Algorithm 1, we iterate over each configuration $j$ of oracle $i$, refining the KNAPSACK oracle to address the Grouped Knapsack Problem and thus arriving at the optimal solution for the optimization problem. The time complexity of the adapted algorithm is asymptotically $O(n_0^2 l/(\delta\Delta))$, where $l$ denotes the number of configurations accessible per oracle.

## F.2. Sandwiched Ensemble Learned Bloom Filters

In sandwiched ELBF, we allocate a BF before each oracle to filter data items before injecting them into the oracle. Suppose we allocate $p_i n$ bits for each pre-filter $i$, whose FPR is then $\alpha^{p_i}$, we can compute the overall FPR as follows:

$$\prod_{i \in \mathcal{O}} \alpha^{p_i} \left( F_{P,i} + (1 - F_{P,i})\alpha^{(b_i - p_i)/F_{N,i}} \right) \tag{28}$$

Thus, the sandwiched ELBF can be optimized by solving the following minimization problem.

$$\min_{\mathcal{A} \subseteq \mathcal{O}} \min_{\mathbf{b}, \mathbf{p}} \quad \prod_{i \in \mathcal{A}} \alpha^{p_i} \left( F_{P,i} + (1 - F_{P,i})\alpha^{(b_i - p_i)/F_{N,i}} \right)$$
$$\text{subject to:} \quad \sum_{i \in \mathcal{A}} (L_i + b_i) \leq T \tag{29}$$
$$b_i \geq p_i \geq 0, \ \forall i \in \mathcal{A}.$$

For each branch $i$, we compute the derivatives with respect to $p_i$ as below:

$$F_{P,i}\alpha^{p_i} \ln \alpha - (1 - F_{P,i})(\frac{1}{F_{N,i}} - 1)\alpha^{b_i/F_{N,i}}\alpha^{(1-\frac{1}{F_{N,i}})p_i} \ln \alpha. \tag{30}$$

The above derivative equals 0 when

$$p_i = b_i - F_{N,i} \log_\alpha \frac{F_{P,i}F_{N,i}}{(1 - F_{P,i})(1 - F_{N,i})}, \tag{31}$$

where we require $F_{P,i} + F_{N,i} < 1$ to ensure $b_i - p_i > 0$ so that the backup BF is not empty.

Thus, the minimum FPR of branch $i$ can be derived as

$$\left(\frac{F_{P,i}}{1 - F_{N,i}}\right)^{1-F_{N,i}} \left(\frac{1 - F_{P,i}}{F_{N,i}}\right)^{F_{N,i}} \alpha^{b_i} \triangleq \Omega_i \cdot \alpha^{b_i}. \tag{32}$$

Consider the inner minimization problem for a given $\mathcal{A}$. We can rewrite the inner minimization problem as below:

$$
\begin{aligned}
\min_{\mathbf{b}} \quad & \sum_{i \in \mathcal{A}} \left( \ln \Omega_i + b_i \ln \alpha \right) \\
\text{subject to} \quad & \sum_{i \in \mathcal{A}} (L_i + b_i) \leq T \\
& b_i \geq F_{N,i} \log_\alpha \frac{F_{P,i} F_{N,i}}{(1 - F_{P,i})(1 - F_{N,i})}, \ \forall i \in \mathcal{A} \\
& F_{P,i} + F_{N,i} < 1, \ \forall i \in \mathcal{A}
\end{aligned}
\tag{33}
$$

By following the same analysis as in Section 3, we can rewrite the outer minimization problem as the following Knapsack problem.

$$
\begin{aligned}
\max_{\mathbf{x}} \quad & \sum_{i \in \mathcal{O}} x_i \cdot \left( \ln \frac{1}{\Omega_i} + L_i \ln \alpha \right) - T \ln \alpha \\
\text{subject to} \quad & \sum_{i \in \mathcal{O}} x_i \cdot \left( L_i + F_{N,i} \log_\alpha \frac{F_{P,i} F_{N,i}}{(1 - F_{P,i})(1 - F_{N,i})} \right) \leq T \\
& F_{P,i} + F_{N,i} < 1, \ \forall i \in \mathcal{O} \\
& x_i \in \{0, 1\}, \ \forall i \in \mathcal{O}
\end{aligned}
\tag{34}
$$

The above problem can be solved by adapting Algorithm 1 after removing oracles with $F_{P,i} + F_{N,i} \geq 1$.

### F.3. Ensemble Learned Bloomier Filter

Bloomier filters (Chazelle et al., 2004) are a variant of BF to solve key-value lookup: each item in $\mathcal{S}$ has an associated value; given an queried item $e$, the corresponding Bloomier filter outputs the value associated with $e$, or $\bot$ if $e \notin S$. We now extend our ELBF to the ensemble learned Bloomier filters by taking the following steps. First, each oracle is trained by regression to output a value or $\bot$ once queried by an input item. If all the oracles output the same value, this value is the final output. Second, we replace the each backup BF in ELBFs by a Bloomier filter to resolve the case where the oracles output inconsistent values. It suffices to record all the items with inconsistent oracle outputs with the corresponding values in the backup Bloomier filters. In this way, our ELBF design can be extended to support key-value lookup.

## G. Additional Experiments

### G.1. Malicious URLs Detection

The main technical challenge in malicious URLs detection (MUD) is to quickly check whether a URL belongs to those flagged as malicious with limited memory overhead, to which BFs offer an efficient probabilistic solution.

In our experiments on the MUD task, we downloaded the dataset from Kaggle, including 1,682,213 URLs, 21.5% of which are malicious, others are benign. 56 numerical features are extracted from these URLs, from which we use 26 features such as URL length, URL unique character ratio, path's length and so on.

In our experiments, we use the same settings for oracles as in the first task of virus signature scan. The size of the large and small oracles are 268Kb and 30-34Kb, respectively. Each small oracles learns from 2 or 3 features.

The experiment results are shown in Figure 6. As in task 1, armed with mutually independent oracles, our proposition ELBF outperforms the state-of-the-art solutions traced in the experiments in terms of the FPR and time overhead. For the correlated case, our ELFF++ has the lowest FPR with less time overhead than Ada-BF and PLBF.

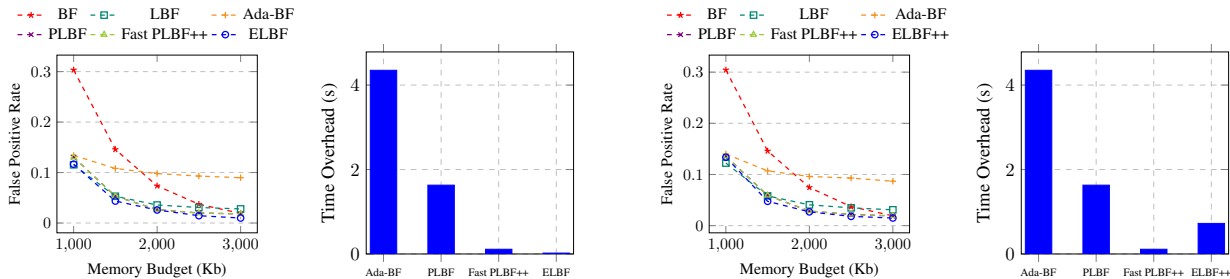

*Figure 6.* Performance comparison: mutually independent case (left), correlated case (right).

### G.2. Universally Unique Identifier Check

In the third task of our experiments, we consider a scenario where the features are independent with each other, by generating 100K Universally Unique Identifiers (UUID). A UUID is a 128-bit number used to uniquely identify an entity. UUIDs are widely used in software systems to avoid maintaining a central naming system. BFs can be used to efficiently check the existence of UUIDs, when dealing with a large volume of identifiers and when a small number of false positives is acceptable.

In our experiments, we build a synthetic dataset with 20 extracted features such as sum of digits, sum of characters, etc., and include the UUIDs whose number and sum of digits are greater than the overall average into the existence. Again, the setting of learning models we use is same as before. The size of the large and small oracles are 267Kb and 34Kb, respectively. Each small oracle learns from 2 features.

The experiment results are shown in Figure 7. Again, our proposition ELBF outperforms the state-of-the-art solutions traced in the experiments in terms of the FPR and time overhead, while ELBF++ outperforms the others in terms of the FPR with a smaller time overhead than Ada-BF and PLBF.

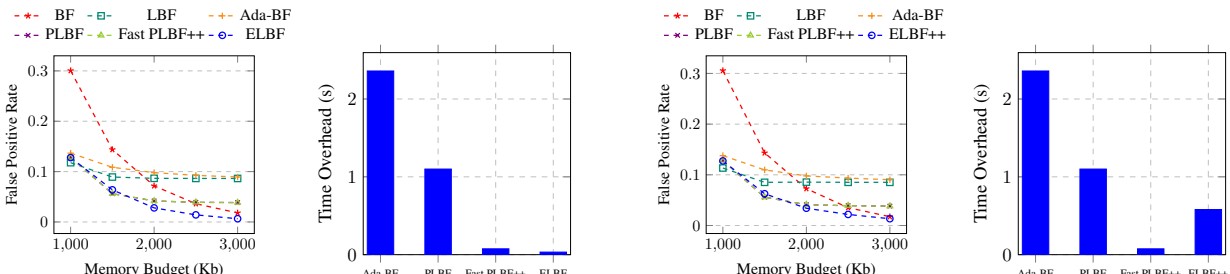

*Figure 7.* Performance comparison: mutually independent case (left), correlated case (right).

### G.3. Time Overhead for Different Sizes of Oracles Pool

We compare the time overhead given a pool of different number of oracles under the space budget of 2000Kb. Again, by time overhead we mean the time to compute the optimal configuration of the entire data structure including the backup filter size, the oracles to use, etc. The experiment results, depicted in Figure 8, show that the time overhead of ELBF is lower than Ada-BF, PLBF and Fast PLBF++ whenever the size of the pool, while that of ELBF++ is lower than Ada-BF and PLBF. We observe that the time overhead of ELBF++ does not increases significantly, as its heuristic search does not traverse all situations practically.

### G.4. Aberration Study w.r.t. Algorithm 1

Given a set of oracles, Algorithm 1 selects the optimal subset. In this subsection, we take a aberration study that compares Algorithm 1 against simpler approaches: (1) RS: random selection, (2) HA: selecting the oracles with the highest accuracy, (3) SM: selecting the oracles with the smallest memory consumption, and (4) HP: selecting the oracles with the highest product of the ratio of accuracy to memory and a factor $\gamma$. Without loss of generality, we default $\gamma$ to 1. For each

compared approach, we optimize the size of backup BF which ensures that each approach fully use the memory budget. The experimental results is shown in Table 2, where we can see that our proposed approach yields the minimal FPR under each budget as expected.

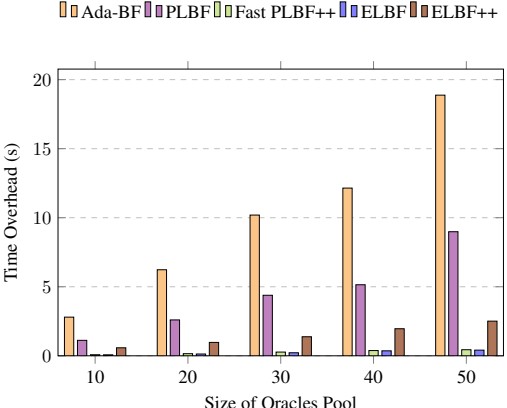

*Figure 8.* Time overhead for different sizes of oracles pool.

*Table 2.* FPR comparison with simpler approaches.

| Budget | RS | HA | SM | HP | Algorithm 1 |
|--------|--------|--------|--------|--------|-------------|
| 1000Kb | 0.1316 | 0.1326 | 0.1251 | 0.1248 | **0.1053** |
| 1500Kb | 0.0706 | 0.0626 | 0.0657 | 0.0655 | **0.0596** |
| 2000Kb | 0.0663 | 0.0352 | 0.0353 | 0.0354 | **0.0333** |
| 2500Kb | 0.0305 | 0.0260 | 0.0258 | 0.0261 | **0.0251** |
| 3000Kb | 0.0202 | 0.0181 | 0.0182 | 0.0179 | **0.0173** |

