# OpenReview forum: "Ensemble Learned Bloom Filters: Two Oracles are Better than One"
_ICML.cc/2025/Conference — ICML 2025 poster_

### Official Review · Reviewer_YvD7 · 2025-03-11

**Overall Recommendation:** 2

**Summary:**

This paper introduces Ensemble Learned Bloom Filters (ELBF), an approach to improving the performance of Learned Bloom Filters (LBF) by leveraging multiple learning oracles of smaller size instead of a single large oracle. The authors formulate the ELBF design as a combinatorial optimization problem: given a pool of oracles and a total space budget, the goal is to select a subset of oracles and determine the sizes of their backup Bloom filters to minimize the overall false positive rate (FPR). The paper draws structural analogies between this problem and the Knapsack problem, developing a Knapsack-based approximate algorithm with proven (ε,δ)-optimality. The authors also propose an extended design, ELBF++, for scenarios with correlated oracles sharing a common backup filter.

**Claims And Evidence:**

The claims made in the submission are generally supported by evidence:

1. The claim that ELBFs outperform standard LBFs under the same space budget is convincingly demonstrated through theoretical analysis and empirical results across multiple datasets.

2. The (ε,δ)-optimality claim for their Knapsack-based algorithm is supported by formal proofs in Theorem 3.2 and Appendix A.

3. The claim about ELBF++ being more effective for correlated oracles is supported by the theoretical analysis in Section 4 and corresponding experimental results.

**Essential References Not Discussed:**

1. Recent work on adaptive Bloom filters that dynamically adjust their parameters
2. Connections to other probabilistic data structures like Count-Min Sketch and their learned variants, which would provide broader context.

**Experimental Designs Or Analyses:**

As mentioned above.

**Methods And Evaluation Criteria:**

I am not familiar with this specific area, so I am unable to evaluate whether the evaluation criteria are appropriate.

**Other Comments Or Suggestions:**

I am not familiar with this specific area, and I will adjust my rating based on the feedback from other reviewers. If other reviewers possess more expertise in this field and provide more technically thorough reviews, their perspectives should take precedence over mine.

**Other Strengths And Weaknesses:**

Strengths:
1. The formulation of ELBF design as a combinatorial optimization problem is elegant and insightful.
2. The theoretical analysis is rigorous, with formal optimality guarantees.
3. The connection to the Knapsack problem provides a solid foundation for the algorithm design.

Weaknesses:
1. The experimental evaluation could include more diverse application domains beyond the specific data analysis tasks presented.
2. The discussion of practical implementation considerations, such as training the multiple oracles efficiently, could be expanded.
3. The hyperparameter sensitivity analysis could be more comprehensive, particularly regarding how to set optimal values in practice.

**Questions For Authors:**

1. How does the performance of ELBF scale with the number of available oracles? Is there a point of diminishing returns, and if so, how can practitioners determine the optimal number of oracles to use? This would help understand the practical limits of the ensemble approach.

2. The paper focuses on minimizing FPR under a fixed space budget. Have you explored other optimization objectives, such as minimizing space usage under a fixed FPR constraint? This alternative formulation might be more relevant for certain applications.

**Relation To Broader Scientific Literature:**

The paper effectively positions itself within the broader literature on:

1. Bloom filters and their variants - Building on the original work by Bloom (1970) and subsequent extensions.

2. Learned Bloom filters - Acknowledging the foundational work by Kraska et al. (2018) and Mitzenmacher (2018).

3. Ensemble learning - Drawing connections to the broader machine learning literature on ensemble methods.

**Theoretical Claims:**

As mentioned above, I am unable to evaluate the correctness of any proofs for the theoretical claims. If possible, I would suggest seeking the expert opinion of other reviewers who are familiar with this specific area.

---

> ### Author Rebuttal · Authors · 2025-03-25
>
> We sincerely thank you for your comments. They are mostly insightful and make us step backwards to rethink several design issues of our algorithm. Please find our response below.
>
> **Answer to Q1.**
> - **Performance of ELBF w.r.t. number of oracles $n_o$:** Theoretically (worst-case complexity bound), in the case of independent oracles, ELBF scales as $O(n_o^2)$; in the case of correlated oracles ELBF++ scales as $O(n_o^3)$; Empirically we observe that the running time scales slightly more linearly in $n_o$. In our opinion, this does not bring significant burden, given that usually the number of available oracles is rather limited.
> - **Number of oracles to use:** There are two parameters/variables here.
>   * If the number of available oracles is fixed, the number of oracles to use as well as which oracles to use are both computed by our algorithms. In fact, when we are provided a pool of oracles, our algorithm computes the set of oracles to use, which implicitly computes the number of oracles to use.
>   * If your comment regards the number of oracles available, our response is that it depends on the quality of the available oracles. We provide the following intuitive analysis. We can define the notion of dominance such that, for a pair of oracles $i$ and $j$, $i$ dominates $j$ if the size of $i$ is not larger than the size of $j$ and the false potive rate of $i$ is not larger than that of $j$ and the false negative rate of $i$ is not larger than that of $j$. Given a pool of oracles, we can first perform pre-processing by removing all the oracles dominated by any other oracle in the pool. This pre-processing operation leaves us with only those oracles that are "pareto" with each other. And it is the number of such pareto oracles that has real impact on the performance of our algorithm. As the parameters of oracles can vary one to another, it is generically difficult to give an answer how many pareto oracles in the pool are "sufficient". To provide a reasonable quantitative response to your question, what we do is as follows: we randomly set oracles with reasonable parameters (in terms of size, false positive and negative rates) in practice and run our algorithm; we observe that in most cases no more than $20$ pareto oracles are enough; beyond this limit, we do not have significant performance gain.
>
> **Answer to Q2.**
>
> This is a very pertinent comment that is typically posed regarding multiple-objective optimization. We have thought about this formulation in our research, which is symmetric to ours. Basically in this formulation we interchange objective and constraint in this formulation. We make two clarifications.
> - The main reason of analyzing the formulation of optimizing FPR by regarding space as constraint is simply because most related work, against which we compare our algorithm, also analyzed this formulation. It is thus fair and easy to perform comparison.
> - The second formulation can be addressed by our algorithm. An intuitive, not necessarily optimal, approach is to gradually increase the space (we discretize the space) and invoke our algorithm to compute the minimal FPR corresponding to the space. Once the FPR reaches the target FPR constraint, we output the corresponding space and the result. This adapted algorithm solves the new formulation of the problem, subject to small error caused by discretization.

---

### Official Review · Reviewer_H5xL · 2025-03-12

**Overall Recommendation:** 4

**Summary:**

This paper studies Learned Bloom Filters (LBF), which enhance traditional Bloom Filters with a learned model (oracle) as a pre-filter. A key challenge in single-oracle LBFs is that the oracle’s size can become a bottleneck when the overall space budget is limited. To address this, the authors propose an ensemble approach that leverages multiple smaller learning oracles and optimizes the associated backup filters. They design and optimize ensemble LBFs for both independent and correlated oracles, demonstrating empirical performance improvements across three practical data analysis tasks. The main technical challenge of the paper is solving the combinatorial optimization problems required for LBF design. For independent oracles, they develop an approximate solution, while for correlated oracles, they propose a greedy heuristic.

**Claims And Evidence:**

The theorems are mathematically proved, and the performance of the proposed algorithms is empirically examined against state-of-the-art baselines.

**Essential References Not Discussed:**

I am not aware of any related works that are essential to understanding the key contributions of the paper but are not currently cited.

**Experimental Designs Or Analyses:**

The experimental setup, baselines, performance measures, and datasets appear reasonable. Moreover, the experimental results align with the theoretical expectations.

**Methods And Evaluation Criteria:**

The proposed algorithms are empirically compared to baselines in terms of memory usage, false positive rate, and time overhead, which are standard evaluation metrics for Bloom filters.

**Other Comments Or Suggestions:**

N/A

**Other Strengths And Weaknesses:**

The idea of combining smaller oracles to overcome memory limitations is intuitive yet interesting and appears effective in practice. This approach offers a novel way to balance model size and accuracy in learned Bloom filters, which could inspire future research in space-efficient data structures.

**Questions For Authors:**

N/A

**Relation To Broader Scientific Literature:**

The paper empirically compares the proposed Ensemble Learned Bloom Filters (ELBFs) with state-of-the-art baselines, including the canonical Bloom Filter (BF) without learning (Bloom, 1970) and several learned Bloom filters: LBF (Kraska et al., 2018), Ada-BF (Dai & Shrivastava, 2020), PLBF (Vaidya et al., 2021), and Fast PLBF++ (Sato & Matsui, 2023). The authors demonstrate improvements over these baselines in almost all experiments.

**Theoretical Claims:**

I did not verify the correctness of the proofs, but the proposed algorithms for the optimization problems are reasonable.

---

> ### Author Rebuttal · Authors · 2025-03-26
>
> We sincerely thank you for the positive feedback. We envision at least two future research directions related to this work: (1) developing theoretically proven algorithms for the correlated case, either exact or approxiation algorithms, ideally with low complexity, (2) extending our idea of orchestrating multiple oracles to other data structure design to improve compactness.

---

### Official Review · Reviewer_xEpC · 2025-03-16

**Overall Recommendation:** 4

**Summary:**

This problem examines whether combining multiple learned oracles can generate a system of lower false positive. In the first case, where each learned oracle is paired with a separate filter, the authors provide theoretical analysis to formulate the problem as a knapsack problem and use dynamic programming to select a configuration minimizing the final false positive rate. In the second case, where a few oracles correspond to one filter, the authors design a greedy algorithm to select the configuration.

update after rebuttal:

I noticed that both ELBF and ELBF++ are evaluated in section 6, so I have removed my previous comment on weaknesses. My score remains unchanged.

**Claims And Evidence:**

The authors provide theoretical analysis and experiments to support their claims.

**Essential References Not Discussed:**

I am only familiar with one previous work Mitzenmacher 2018, which is listed in the paper, so the discussion of related work looks good to me.

**Experimental Designs Or Analyses:**

Yes, the experiment setup in section 6 looks good.

**Methods And Evaluation Criteria:**

The experimental part is done by comparing false positive rate under limited memory budget, which makes sense to me.

**Other Comments Or Suggestions:**

no

**Other Strengths And Weaknesses:**

Strength:
1. The idea of ensembling learned bloom filters seems to be new to me and the idea is supported by both theoretical and empirical evidence.
2. The theoretical analysis which uses lagrange multiplier to transform the configuration selection into a knapsack problem seems very nice and clean.
3. The proposed method is compared with many other baselines in a very straightforward experiment setup.

Weakness:
1. It seems that there is no theoretical guarantee for the correlated oracle case

**Questions For Authors:**

I wonder why the authors don't perform experiments to test their algorithm in section 4, as I understand that it may be too hard to derive theoretical guarantees for their greedy algorithm.

**Relation To Broader Scientific Literature:**

n/a

**Theoretical Claims:**

I have checked the theoretical proof in section 3, which looks good.

---

> ### Author Rebuttal · Authors · 2025-03-26
>
> We sincerely thank you for the positive feedback. Our greedy algorithm developed in Section 4 is tested and evaluated in our primary experiments in Section 6 on Experiments, where it outperforms baselines under various memory constraints. More in-depth experiments are presented in Appendix due to page limit for the main text.

---

### Decision · Program_Chairs · 2025-05-01

**Decision:**

Accept (poster)

**Comment:**

The paper introduces Ensemble Learned Bloom Filters (ELBF), a framework that improves the efficiency of learned Bloom filters (LBF) by replacing a single large oracle with a combination of multiple smaller learned oracles. The authors formulate the ELBF design problem as a combinatorial optimization task and solve it using a dynamic programming approach based on the Knapsack problem for independent oracles, and a greedy heuristic for correlated oracles (ELBF++). The goal is to minimize the false positive rate (FPR) under a fixed space constraint. The authors support their claims with theoretical guarantees and comprehensive experiments across several benchmark tasks.

Most of the reviewers recommended acceptance. The reviewers appreciated the novelty of the solution, theoretical insights and experimental results.  They also suggested expanding the discussion on training multiple oracles, hyperparameter tuning, and practical deployment consideration, as well as expanding related work section. The authors should take this into account when preparing the final version of the paper.